# Developing an AI-assisted clinical decision support system to enhance in-patient holistic health care

**Wang-Chuan Juang**[1,2]*, **Ming-Hsia Hsu**[3,4], **Zheng-Xun Cai**[4], **Chia-Mei Chen**[4]*

**1** Quality Management Center, Kaohsiung Veterans General Hospital, Kaohsiung, Taiwan, **2** Department of Business Management, National Sun Yat-sen University, Kaohsiung, Taiwan, **3** Department of Information Management, Kaohsiung Veterans General Hospital, Kaohsiung, Taiwan, **4** Department of Information Management, National Sun Yat-sen University, Kaohsiung Taiwan

* wcchuang@vghks.gov.tw (WCJ); cmchen@mis.nsysu.edu.tw (CMC)

**Data Availability Statement:** Data cannot be shared publicly due to the data sensitivity and the data sharing regulation. Data are available from the Kaohsiung Veterans General Hospital Ethics Committee (contact via 886-7-342-2121) for

## Abstract

Holistic health care (HHC) is a synonym for complete patient care, and as such an efficient clinical decision support system (CDSS) for HHC is critical to support the judgement of physician's decision in response of patient's physical, emotional, social, economic, and spiritual needs. The field of artificial intelligence (AI) has evolved considerably in the past decades and many AI applications have been deployed in various contexts. Therefore, this study aims to propose an AI-assisted CDSS model that predicts patients in need of HHC and applies an improved recurrent neural network (RNN) model, long short-term memory (LSTM) for the prediction. The data sources include in-patient's comorbidity status and daily vital sign attributes such as blood pressure, heart rate, oxygen prescription, etc. A two-year dataset consisting of 121 thousand anonymized patient cases with 890 thousand physiological medical records was obtained from a medical center in Taiwan for system evaluation. Comparing with the rule-based expert system, the proposed AI-assisted CDSS improves sensitivity from 26.44% to 80.84% and specificity from 99.23% to 99.95%. The experimental results demonstrate that an AI-assisted CDSS could efficiently predict HHC patients.

## Introduction

Holistic health care (HHC) is to achieve comprehensive patient care from different aspects, including physical, emotional, social, economic, and spiritual needs. It provides an in-depth understanding of patients and their various care needs [1]. There are different ways of implementing HHC services [1–4]. The HHC defined in this study is to provide multidisciplinary patient care for the patients in critical conditions, where the multidisciplinary care requires services from multidisciplinary medical professions, such as physicians, therapists, nurses, dietitians, psychologists, pharmacists, rehabilitation care specialists, and social workers, in order to address patients' physical, emotional, social and spiritual needs and to improve their lives [5]. Non-HHC patients mainly receive treatments from one medical department.

researchers who meet the criteria for access to confidential data.

**Funding:** This research is funded by Kaohsiung Veterans General Hospital (Grant ID: KSVNSU110-012). The funder had no role in study design, data collection and analysis, decision to publish, or preparation of the manuscript. The funder provided support in the form of salaries for Juang and Hsu, but did not have any additional role in the study design, data collection and analysis, decision to publish, or preparation of the manuscript. The specific roles of these authors are articulated in the 'Author Contributions' section.

**Competing interests:** The authors have declared that no competing interests exist.

The world is aging at a rapid rate and by 2030 there will be 34 super-aged nations where more than 20% of the population is over 65 [6]. This issue has broader implications for health care needs and demanding medical resources. On the other hand, as the number of patients with multiple comorbidities grows, so does the need for comprehensive health care services. The elderly with multiple diseases requires treatments across multidisciplinary medical professionals and rely on an efficient patient-centered holistic care system that keeps track of all the medical encounter records for monitoring disease progress, improving patient satisfaction, and reducing health care cost.

With limited resources, hospitals and medical staff need an efficient clinical decision support system to forecast holistic care needs to improve medical quality, staff work efficiency, and patient satisfaction. The field of AI has evolved considerably in the last decades and many AI applications have been deployed in various contexts. However, it remained challenging to apply AI technology for forecasting patients in need of holistic care. This study aims to develop an AI-assisted model that identifies patients in need of HHC.

AI is a big computer science field that encompasses logic, probability, and continuous mathematics and performs sophisticated tasks in different industries. AI includes but is not limited to machine learning (ML), deep learning (DL), and natural language processing (NLP), and whichever techniques can help medical staff find important information from data. Some medical organizations already applied AI for daily routines as a decision support system, it can assist medical staff to make the decision, reduce the false rate and increase productivity by performing risk assessments, analyzing different types of diseases, and finding the relationships among the medical data.

A review [7] surveyed the literature on ML-based CDSS and concluded that comprehensive patient data may improve the ability of ML-based CDSS. Another review [8] assessed the performance of the ML algorithms on prediction and classification, including neural networks (NN), logistic regression (LR), and support vector machines (SVM). The results showed that NN algorithms outperform others significantly and that the ML technology predicts outcomes accurately. One performance study [9] investigated SVM, LSTM, and Autoregressive integrated moving average (ARIMA) on predicting the hepatitis E disease and demonstrated that LSTM yields the best performance in all the performance metrics. Another study [10] conducted by Google experimented over ten thousand different RNN architectures and concluded that LSTM outperforms RNN. Based on the literature review, LSTM outperforms many contemporary ML models including LR, SVM, NN, RNN, and ARIMA. Therefore, this study adopts LSTM to develop the proposed prediction model.

A study [11] outlined a process of building a ML model for healthcare applications, where the process consists of the following phases: problem selection, data collection, ML development, model validation, impact assessment, and deployment and monitoring. ML promises the avoidance of biases in diagnosis and treatment, as the algorithms could objectively interpret the medical data. Integrating ML with CDSS may improve clinical decisions. However, as ML models learn from historically collected data, two studies [12, 13] pointed out that the ML algorithms may be subject to biases for missing data and imbalanced data. Therefore, an efficient ML-based CDSS should address the above data biases issues. This study employs interpolation to avoid the bias caused by missing data and applies data resampling to mitigate data imbalance problem.

Yahyaoui et al. [14] proposed a ML-CDSS for diabetics prediction and compared conventional ML with DL approaches. Their evaluation showed that random forest was more effective than Convolutional Neural Network (CNN) and SVM. Elani et al. [15] applied an ML model to predict tooth loss and suggested that models incorporating additional socioeconomic characteristics perform better than those relying on routine dental care records.

A study [16] applied AI to mental health care. Patients are monitored, and their daily information is analyzed by AI to inspect if they are in good mental health. Frangou et al. [17] installed a microelectronic sensor on the cap of a pill bottle to monitor the medication adherence of patients. The sensor records and transmits the timestamps of opening the bottle, and the clinician checked if the patient's medication adherence is lower than 50%. The experimental results indicate that the system succeeded to monitor patients' medication adherence.

Liu et al. [18] developed a CNN model to identify tuberculosis infection from X-ray images. They applied a data set of 4701 X-ray images, 453 normal and 4249 abnormal, to evaluate the proposed model and it yielded accuracy of 85.68%. Another study also [19] applied a CNN model on chest X-ray image classification, but it detected multiple diseases such as pleural thickening, otosclerosis, and pulmonary interstitial hyperplasia, not just tuberculosis. This multi-classification model required more than 16 thousand labeled X-ray images to train and yielded accuracy of 82.2%.

A study [20] reviewed computerized CDSS to support automatic detection of critical conditions and observed a trend toward the use of data-driven algorithms. Given the current lack of best practice guidance to AI, a study [21] highlighted that patients and healthcare professionals require clinical prediction models to accurately guide healthcare decisions and that AI models potentially improve diagnostic accuracy and reliable prediction. A study [22] commented that the adoption of emergent technologies lags far behind and called for healthcare professionals and institutes to get prepared for adopting these new technologies. A study [23] applied NLP techniques to extract epilepsy data from clinic letters. A study [24] conducted a review on AI health-related applications in low- and mid-income countries and concluded that data from these countries were used for AI research. An expert system integrated with a neural network model, and production rules [25] were proposed for stroke diagnosis and treatment plan provision.

## Materials and methods

### Patient consent for publication

Not applicable. This research was conducted with fully anonymized and de-identified patient records and did not cause any medical interference to patients.

### Ethics approval

This research has received ethical approval from the IRB of Kaohsiung Veterans General Hospital with the IRB certification number KSVGH20-CT10-08.

### Holistic healthcare platform

An HHC platform has been implemented in the hospital in 2017, which integrates multiple healthcare systems used by clinic staff and builds up a communication channel among multi-disciplinary clinic staff. The HHC service is initiated when the physician who takes charge of a patient's treatment decides to. They all make the judgment based on the diagnoses and patient's daily conditions recorded on the HHC platform. Fig 1 outlines the HHC procedure.

Fig 2 illustrates the main user interface of the platform. The interface is in a grid format, where each grid links to a specific healthcare information system in order to provide further detailed information in support of the service. The upper-left grid links to the patient care information system showing electronic medical records, the mid-left grid connects to the nursing information system, the lower-left grid shows the status of multi-disciplinary diagnosis, the upper-right grid serves as a handover communication channel for HHC staff, the mid-

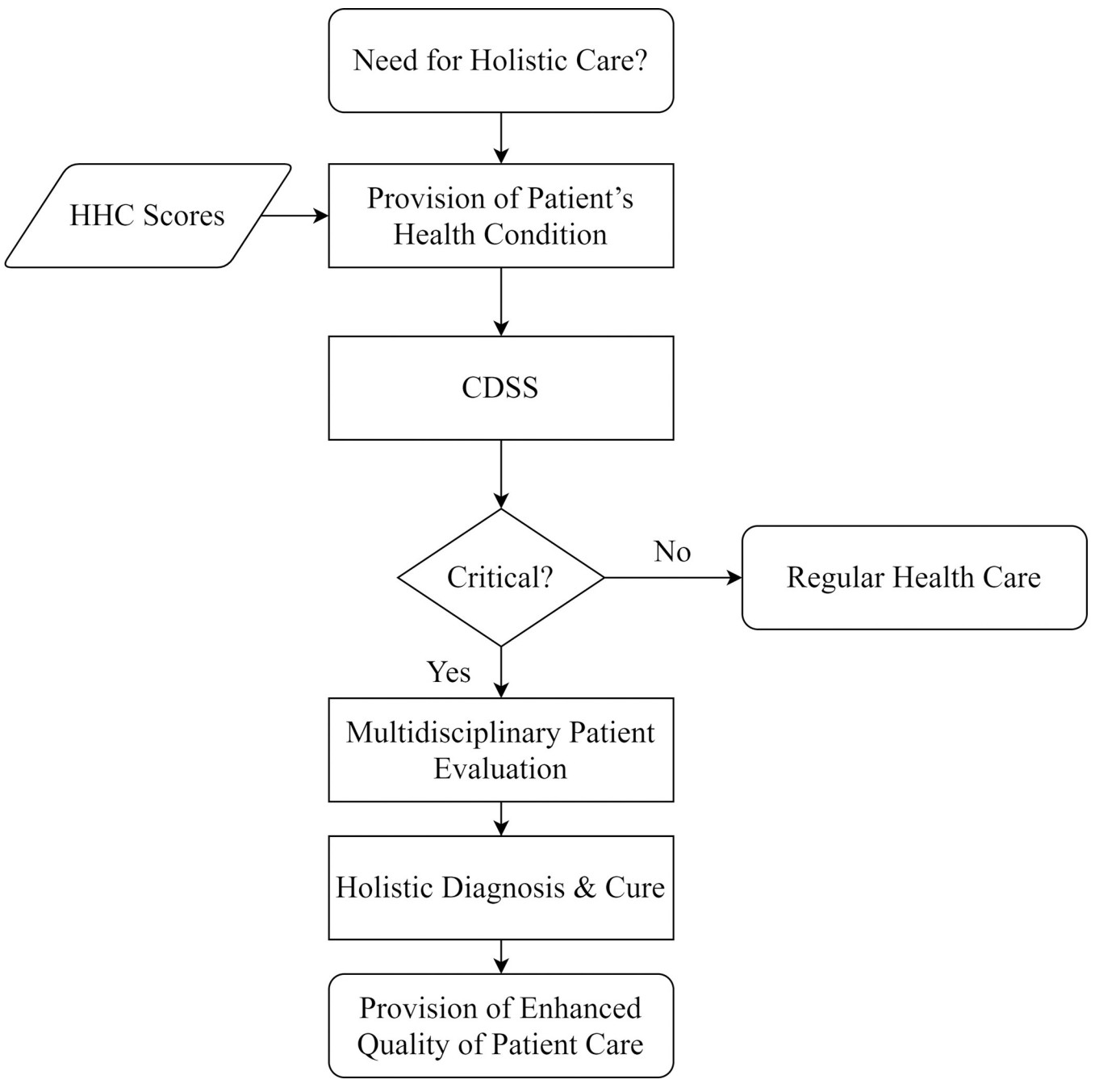

**Fig 1. HHC flow chart.**

right grid monitors the patients with high or anomalous early warning score, and the lower-right grid compiles the care log records from multidisciplinary clinic staff, including therapists, pharmacists, social workers, physician assistants, hospice care specialists, rehabilitation care specialists.

In this platform, an HHC multidisciplinary diagnosis is made by a team of clinic specialists who take charge of a specific holistic healthcare patient. To reduce physician's workload, a rule-based expert system has been deployed when the hospital started to provide the HHC

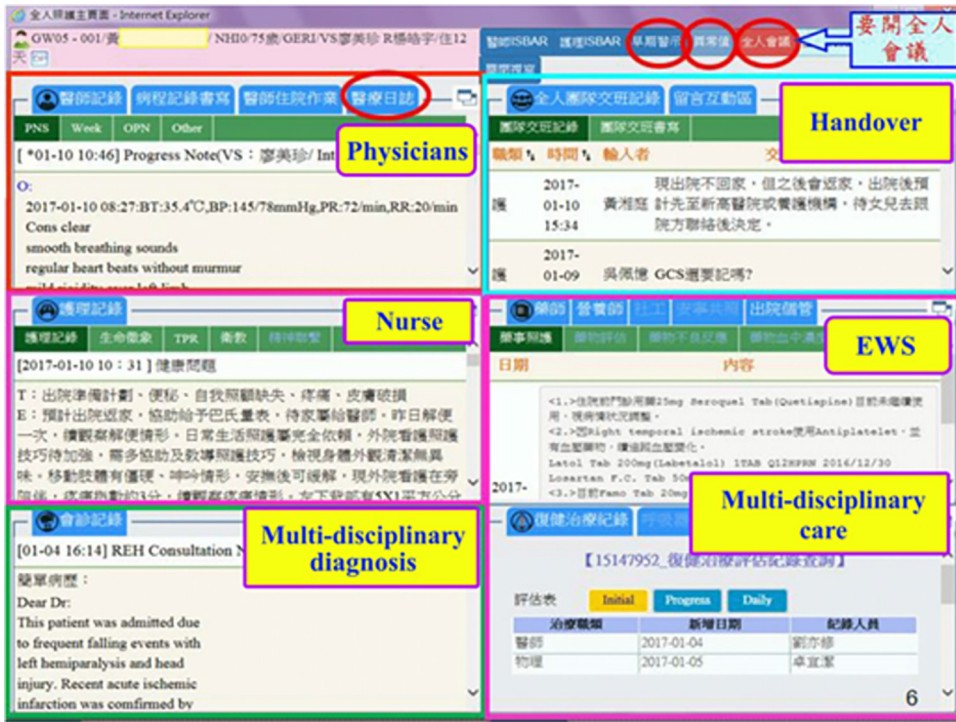

**Fig 2. The main user interface of the HHC platform.**

service. The rules in the CDSS are defined by a group of physicians from multiple departments, which attempt to mimic the experts make HHC decisions. However, the rule-based CDSS could not predict HHC efficiently as expected, which motivates this study to improve the prediction performance. The prediction performance in this study is calculated based on the clinicians' decisions.

## Study population

This study adopted de-identified in-patient medical records collected over two years from a tertiary medical center to build and evaluate the proposed method. This study acquires the medical data from the aforementioned HHC platform, where the physiological health conditions of all hospitalized patients are available in the Web-based EMR, including vital signs, emotional conditions, sleeping quality, food intake, medication status, etc. A total of 121 thousand in-patient cases and 890 thousand medical records were referenced dedicated to training and evaluation, where the dataset was collected from 2017 to 2019 after the holistic care service has been deployed in the hospital and the HHC patients were verified manually by physicians. Cross-validation tests were performed to eliminate the overfitting issue, where different portions of training and testing were evaluated.

## AI-assisted CDSS

Fig 3 presents the proposed system architecture, where the left dashed box indicates the process of building an ML-based prediction model based on the historical data and the right box shows the process of applying the model to support HHC decision making. The training data comprises historical medical data collected during the past period that the hospital provided in-patient HHC services for research approved by IRB, which included daily medical

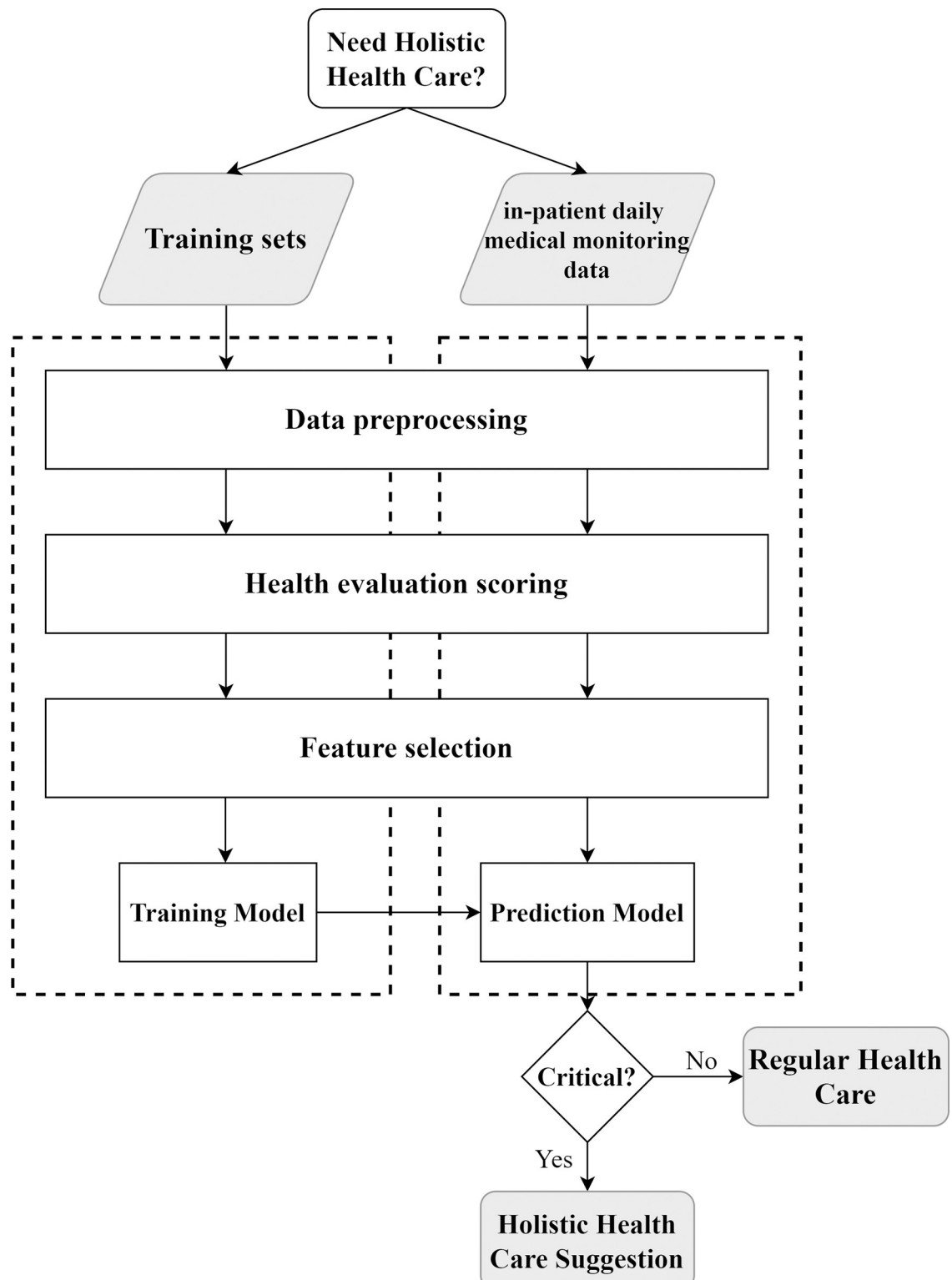

**Fig 3. The proposed holistic health care decision support system architecture.**

**Table 1. Patient health condition attributes.**

| TeamEval | | CCI | | EWS | |
|---|---|---|---|---|---|
| Index | Ratio | Index | Ratio | Index | Ratio |
| Pain | $R_{Pain} = 5$ | Min(B, 20) | $R_{CCI} = 5$ | Min(C, 20) | $R_{EWS} = 5$ |
| Brief Symptom Rating Scale (BSRS) | $R_{BSRS} = 5$ | | | | |
| Nutrition | $R_{Nutrition} = 30$ | | | | |

records extracted from in-patient care EMR and HHC status in all admitted patients, whether patients were chosen for HHC or not. To employ an ML algorithm to assist the HHC decision support, the data was cleaned to remove errors in order to improve the accuracy of the ML model. After data preprocessing, module Health Evaluation Scoring quantifies the patient's daily health status. Module Feature Selection extracts relevant medical information and transforms each selected feature into a form that can represent the meaning of the data for the ML model.

The module Data Preprocessing examines the validity of the patient's records, including missing data points, empty values, and outlier values. It applies linear interpolation to estimate missing data, for example, a missing blood pressure measurement in a patient's daily vital record. A default value 0 is filled in for null or empty value in a data field. A value larger than 2 standard deviations of a given field is considered as an outlier and is replaced by an interpolation value.

The module Health Evaluation Scoring develops a health score to evaluate a patient's total health condition by weighting for each health index to quantify the degree of the HHC need. The patient health condition is defined from the following three aspects: patient's wellness evaluation from the care team (TeamEval), comorbidity condition based on Charlson Comorbidity Index (CCI) [26], and Early Warning Score (EWS) [27] as summarized in Table 1, where Pain and BSRS range from 0 to 10, Nutrition ranges from 0 to 6, CCI scales 1 to 6, and EWS scales 0 to 3. Each index in Table 1 contributes a ratio tuned by the experts, where B stands for the measured CCI value and C for the EWS value. Different diseases contribute different mortality risks to a patient, so weighting is given for the above three wellness aspects as listed in Table 2. The Health Score is defined below.

$$HealthScore = TeamEval \times W_{TeamEval} + CCI \times W_{CCI} + EWS \times W_{EWS} \qquad (1)$$

$$\text{where } TeamEval = Pain \times R_{Pain} + BSRS \times R_{BSRS} + Nutrition \times R_{Nutrition} \qquad (2)$$

Table 3 lists the selected features in the proposed AI-assisted model, where the data types are classified into three types: continuous numerical, discrete numerical, and categorical. Each type requires different encoding methods in order to feed into the AI model.

**Table 2. Weighting for different diseases.**

| Disease | $W_{TeamEval}$ | $W_{CCI}$ | $W_{EWS}$ |
|---|---|---|---|
| Cerebrovascular accident (CVA) | 0.37 | 0.34 | 0.29 |
| Dementia | 0.7 | 0.19 | 0.10 |
| Cancer | 0.61 | 0.19 | 0.20 |
| Others | 0.10 | 0.80 | 0.10 |

For a continuous numerical feature, for example, temperature or blood pressure, the proposed system normalized the data in order to quantify the significance of the feature value. This study adopted MinMaxScaler standardization approach that transformed the data by scaling it to a given range to preserves the shape of the original distribution. For each value in a feature X, MinMaxScaler subtracts the minimum value in the feature and divides by the range, where the range is the difference between the original maximum and the original minimum of the feature. MinMaxScaler standardization is expressed in the following formula, where $max$ and $min$ stand for the maximum and minimum values of feature $X$, respectively, and $X_{std}$ is the standardized value of feature $X$ ranging in 0~1.

$$X_{std} = \frac{X - min}{max - min} \tag{3}$$

Regarding discrete numerical features, such as check-in and check-out dates, this study transformed these parameters into meaningful values for the ML model, for example in this case the number of hospitalized days. The number of hospitalized days, $D_{delta}$, a discrete numerical feature, can be expressed as follows, where $DD$ and $ED$ mean the check-out and check-in dates, respectively.

$$D_{delta} = DD - ED \tag{4}$$

A discrete numerical feature, such as Coma Status, combines multiple pieces of medical information into one. The Glasgow Coma Scale (GCS) is a neurological scale to give a reliable and objective way of recording the state of a patient's consciousness, which contains three physical aspects of coma seriousness: eye ($CS_{eye}$), vocal ($CS_{vocal}$), and motion ($CS_{motion}$). This study combines the above three scales into one feature, Coma Status ($CS_{all}$), as expressed below.

$$CS_{all} = CS_{eye} + CS_{vocal} + CS_{motion} \in [1, 15] \tag{5}$$

The Early Warning Score (EWS) quantifies the physiological measurements routinely recorded at the patient's bedside, which is an important attribute for identifying acutely ill patients. Table 4 summarizes the EWS scoring adopted in this study.

Regarding categorical data, such as disease or gender, it is encoded into a bit string where each bit represented one element in the set in this study.

**Table 3. The selected features.**

| Feature Type | Feature name | Description |
|---|---|---|
| Continuous numerical | Age | The patient's age. |
| | Health Score | It is calculated in the module Health Evaluation Scoring. |
| | Medical information | The medical information includes daily body temperature, heart rate, respiration rate, oxygen prescription. |
| Discrete numerical | Hospitalization days | The number of days that the patient resides in the hospital. |
| | Early Warning Score | The EWS is calculated based on daily medical records, where Table 4 outlines the scoring and EWS is a positive integer ranging from 0~21. |
| | Coma Status | The Coma Status sums up the three coma indicators of consciousness: eye reaction, vocal reaction, and motion reaction, and is a positive integer ranging 1~15. |
| Categorical | In-patients ID | The patient's pseudonymized identifier. |
| | Hospitalization date | The timestamp of hospitalization. |
| | Disease | The disease indicates the illnesses that the patient has. |

**Table 4. Early warning score.**

| Physiological parameters | Value | | | | | | |
|---|---|---|---|---|---|---|---|
|  | 3 | 2 | 1 | 0 | 1 | 2 | 3 |
| Respiratory rate | < = 8 |  | 9–11 | 12–20 |  | 21–24 | > = 25 |
| Oxygen saturation | < = 91 | 92–93 | 94–95 | > = 96 |  |  |  |
| Oxygen prescription |  | YES |  |  |  |  |  |
| Body temperature | < = 35 |  | 35.1–36.0 | 36.1–38.0 | 38.1–39.0 | > = 39.1 |  |
| Systolic blood pressure | < = 90 | 91–100 | 101–110 | 111–219 |  |  | > = 220 |
| Heart rate | < = 40 |  | 41–50 | 51–90 | 91–110 | 111–130 | > = 131 |
| Coma Status |  |  |  | > = 15 |  |  | 1–14 |

Long Short-Term Memory (LSTM) is an improved RNN model that learns the temporal dependence of a sequence and predicts the next one, so it is suitable for learning the changes of a patient's health status in a time frame and predicting HHC needs. Fig 4 presents an illustration of the proposed LSTM model composed of three layers, where the input layer is fed with the time-series data: the current medical data ($D_t$) plus past 7-day patient's medical records ($D_{t-7} \sim D_{t-1}$) and the output layer forecasts if the given patient needs HHC.

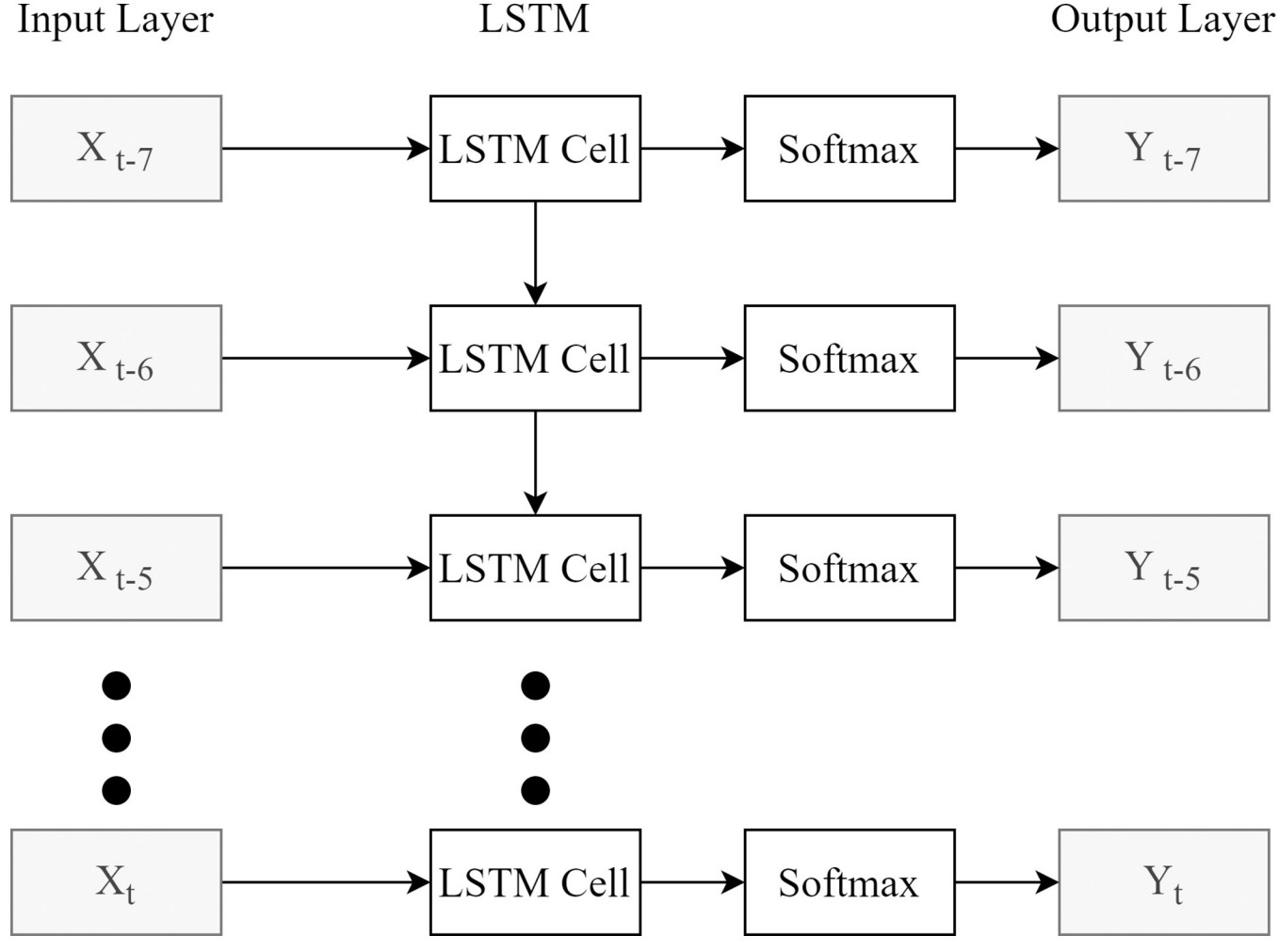

**Fig 4. The proposed LSTM prediction model.**

### Rule-based CDSS

A group of experts with domain knowledge designed a set of rules that define patient's need for HHC. The rules have been refined by the experts over a long period of time, and the parameters and weightings were experimented with and reviewed by clinic staff several times. The rule-based CDSS suggests an HHC service if one of the following rules is satisfied:

1. HealthScore $\geqq$70

2. HealthScore $\geqq$50 & Nutrient $\geqq$2 & BSRS $\geqq$4

3. Dementia's HealthScore $\geqq$40 & exclude REH patient

4. Nutrition $\geqq$2 & BSRS $\geqq$4 & Pain $\geqq$3

## System evaluation

### Evaluation dataset

The evaluation data was collected after the HHC service has been deployed. The evaluation dataset excludes the following cases: (a) psychiatric in-patients; (b) cases for demo purpose during its initial deployment phase; (c) patients who have been hospitalized at least 7 days; (d) cases that received HHC within 7 days after admission. The latter two types of cases were excluded, as there are not enough history records to justify if they are needed or not. After the aforementioned data cleaning, the evaluation dataset contains 552 HHC cases, 121 thousand in-patients, and 890 thousand medical records. The characteristics of the dataset are summarized in Table 5.

### Patient and public involvement

This research was carried out without patient or public involvement in the design or result interpretation. Patients and members of the public did not contribute to the writing or editing of this manuscript.

**Table 5. The characteristics of the dataset.**

|  | Mean ± standard deviation | Range |
|---|---|---|
| Age | 53.79 ± 24.14 | 0~107 |
| Hospitalization days ($D_{delta}$) | 7.89 ± 11.59 | 1~1331 |
| Nutrition | 1.72 ± 0.85 | 1~6 |
| Brief Symptom Rating Scale (BSRS) | 1.83 ± 1.01 | 1~10 |
| Pain | 1.93 ± 1.02 | 1~10 |
| CCI | 4.69 ± 2.37 | 0~19 |
| EWS | 3.85 ± 2.81 | 0~20 |
| Respiratory rate (breaths/min) | 18.08 ± 2.34 | 0~50 |
| SpO2 (%) | 96.91 ± 2.27 | 80~100 |
| Systolic blood pressure (mmHg) | 128.46 ± 21.06 | 0~203 |
| Pulse | 81.63 ± 15.79 | 0~133 |
| Temperature (C) | 36.41 ± 0.59 | 32.0~42.5 |
| $CS_{eye}$ | 3.88 ± 0.46 | 1~4 |
| $CS_{vocal}$ | 4.36 ± 1.33 | 1~5 |
| $CS_{motion}$ | 5.74 ± 0.79 | 1~6 |

**Table 6. The confusion matrix.**

| | | LSTM (proposed) | | RNN | | Rule-based | |
|---|---|---|---|---|---|---|---|
| Real / Prediction | | True | True | False | False | True | False |
| True | | 211 | 132 | 201 | 195 | 69 | 466 |
| False | | 50 | 60027 | 60 | 59964 | 192 | 59693 |

## Analysis and statistical tests

Most AI models work well when the number of samples in each class is about equal. However, this dataset is highly imbalanced with only 0.43% of HHC cases, so equal-sized classes by either under-sampling or over-sampling might affect the training performance. To train a good prediction model, this study applies under-sampling the majority class to produce the training dataset of a ratio of 1:4 (HHC: non-HHC). The HHC cases are divided into two equal-sized sets for training and testing.

In order to improve the performance of the prediction models, the hyperparameter tuning was performed with 10-fold cross-validation, where the class imbalance is taken into account in the parameters of the models. Given that the evaluation dataset exhibits target imbalance, the trained classifier is imbalance-aware so that it would not skew predictions towards the majority class.

This study adopts cross-validation to evaluate the system performance. A prediction is considered valid if the forecasting timestamp of an HHC falls within two weeks of the actual date of occurrence. In order to assess the goodness of the prediction model, this study estimates accuracy, sensitivity, specificity, and area under the curve of the receiving operating curve (ROC-AUC). Accuracy is the proportion of the correct predictions to the total cases; sensitivity (recall or true positive rate) is the fraction of the correct HHC predictions over the HHC predictions; specificity (true negative rate) is the fraction of the correct non-HHC predictions over the non-HHC predictions.

High accuracy means the model identifies a majority of HHC and non-HHC patients correctly; high sensitivity means the model identifies a majority of HHC patients correctly; high specificity means the model identifies a majority of non-HHC correctly. For the use case presented, sensitivity is more important than precision. The consequence of missing patients who need HHC is worse than over-calling potential HHC patients.

## Results

Several past medical studies applied RNN for classification. Therefore, this study conducts experiments and compares the proposed LSTM model against the rule-based expert system and RNN model. Table 6 summarizes the performance confusion matrix; Table 7 lists the performance measures; and Fig 5 outlines the ROC curve of the proposed method.

**Table 7. The results of the performance comparison.**

| | LSTM (proposed) | RNN | Rule-based |
|---|---|---|---|
| Accuracy | **99.65%** | 99.57% | 98.91% |
| Specificity (NPV) | **99.95%** | 99.68% | 99.23% |
| Sensitivity (Recall) | **80.84%** | 77.33% | 26.44% |
| Precision (PPV) | **61.52%** | 50.76% | 12.90% |
| F1 Score | **0.6987** | 0.6119 | 0.1734 |

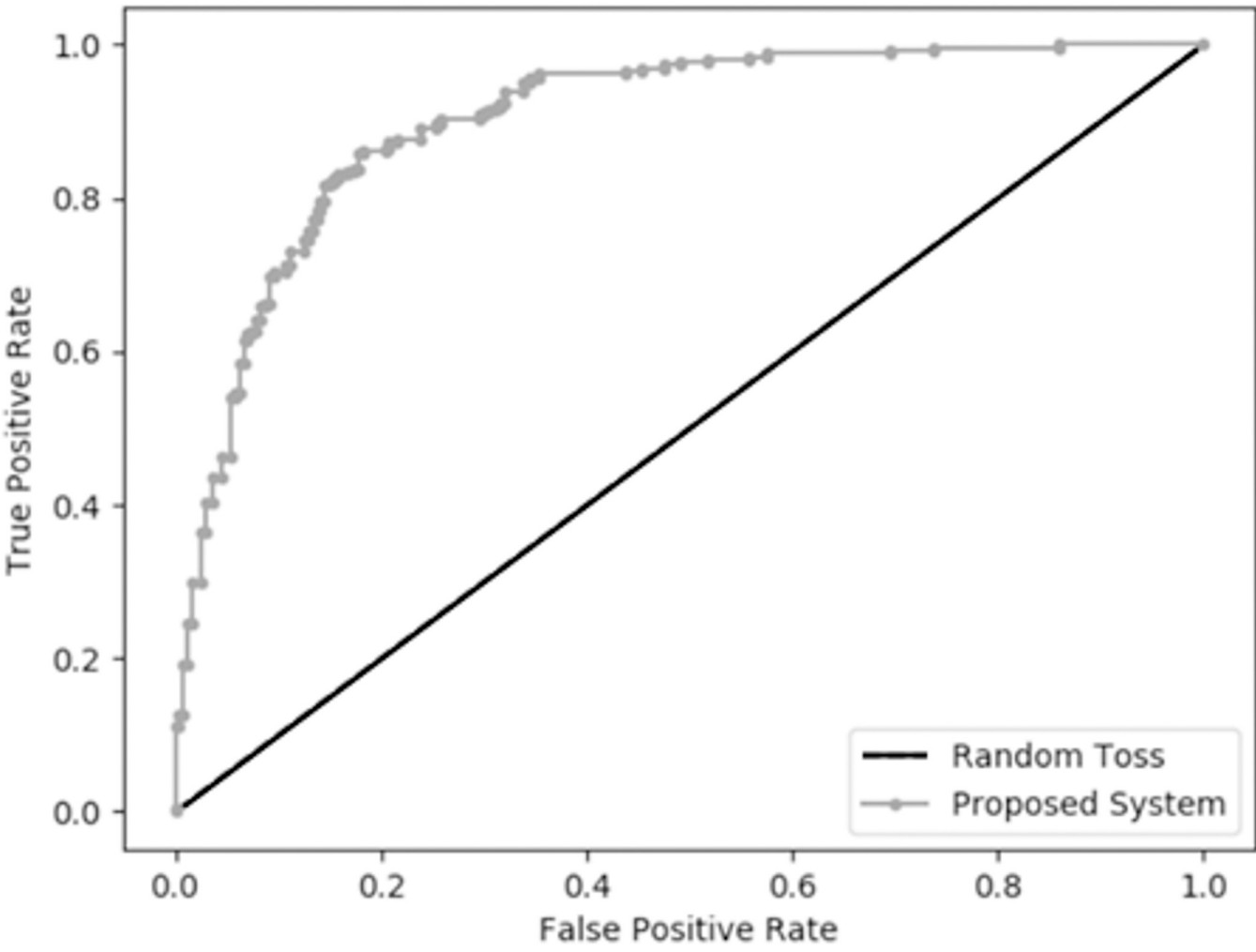

**Fig 5. The ROC curve of the proposed LSTM method.**

## Discussion

This study found that the AI algorithm constructed with neural networks from patient's daily medical data enables HHC prediction and the AI-assisted CDSS outperforms the expert-defined rule-based approach. The proposed LSTM model predicts HHC and non-HHC efficiently and yields 99.65% accuracy, 80.84% sensitivity, and 99.95% specificity, while the rule-based solution could not identify HHC correctly and yields low sensitivity of 26.44%. The ROC curve of the proposed method is above the curve of random toss, and the AUC = 0.903 demonstrates that the proposed method performs very well on prediction. High sensitivity is important for the proposed CDSS as it identifies HHC patients efficiently and achieves the purpose of improvement of healthcare quality.

AI is bringing a paradigm shift to healthcare, powered by increasing the availability of healthcare data and advanced analytic techniques. To our best knowledge, the present study is the first attempt to apply an AI model to develop a CDSS in support of HHC. This is also the first attempt to demonstrate that an LSTM-based CDSS outperforms the rule-based expert system.

Furthermore, this study compares the performance of two different AI models: LSTM and RNN. There are only a few published reports focusing on applying AI for CDSS or HHC, and most discussed the impacts and needs of applying AI to healthcare. This study developed an innovative AI-assisted CDSS predicting HHC patients based on their daily physiological medical records and historical medical information.

An intuitive and traditional approach for CDSS defines rules and applies thresholds to determine whether a patient needs HHC. The rules defined by domain knowledge experts in the rule-based method are limited. The inter-dependency relationship among the medical indicators of a patient's health condition is so complicated that human-defined rules might not be able to comprehend.

The rule-based approach has been practiced by the studied hospital for two years, and its inefficiency disrupted the usability of the HHC service. The rule-based approach did not consider the progress of the patient's condition over a period of time and missed some HHC cases. Our preliminary study investigated the missed (false negative) cases, where the patient's situation got worse but has not been able to trigger the rules. The proposed LSTM model could identify the aforementioned situation, as it learns efficiently temporal dependency of time-series data and predicts HHC based on the patient's past health condition.

The experimental results indicate that both AI models (RNN and LSTM) perform better than the rule-based approach and imply that AI is a promising solution for improving medical information systems and assisting clinical judgment. The strong point of this present study is that the proposed LSTM algorithm could recognize hidden features from the medical records that were clinic essential parameters for predicting HHC. The LSTM is an improved version of RNN, which is effective in learning time-series data. The performance comparison results indicate that LSTM performs better than RNN.

Many factors affect the success of AI on a given task, and among them, the representation and quality of data are critical. A study [28] indicates that irrelevant, redundant, noisy, or unreliable data makes knowledge discovery difficult during the training phase. Therefore, data preprocessing and feature selection are important for constructing an effective prediction model. This study reflects their findings and demonstrates that encoding features well could build an efficient AI model.

## Strengths

This study adopted a gold-standard dataset of de-identified clinic data accepted by clinical staff as an accurate and reliable reference to establish the proposed system which provides effective suggestions to predict HHC needs. The proposed system was evaluated using medical records from a hospital that supports HHC service and the expected results were verified by experienced clinicians. This study implemented the proposed system in Python and applied the open-source AI library Scikit-learn to build up the proposed AI model. The library supports TensorFlow and can improve the efficiency of executing AI algorithms. The proposed approach provides a useful model for the development of similar CDSS applications in the future.

## Weaknesses and limitations

Our findings should be interpreted in light of the following limitations. The current analysis was performed in a single hospital in Taiwan, so the selection of patients may be biased. This limited the number of patient diversity available to validate our algorithm, and therefore, the generalizability of the proposed algorithm may be limited. Further validation analyses using other datasets are necessary to establish the validity of the proposed approach.

HHC service implementation may vary in other hospitals, which might affect the key features and prediction performance. However, this study has made efforts to extract features from patient medical records commonly provided by all clinics rather than relying on specific data from the target hospital.

It is difficult to account for the variability of the disease risks to a new illness such as COVID19, which may generate a new situation that needs HHC and defines new time-series patterns of patient's health conditions. The AI approach needs to be more adaptive to learn such new patterns. Further work could be focused on employing reinforcement learning models to improve the flexibility of the prediction model. However, this would require a significant amount of time to acquire and analyze the decisions made by clinicians and the implications of a new disease to HHC patients.

## Conclusion

Quality of care for patients with comorbidity is one of the most important and complex problems of modern clinical health care because they can lead to patient incapacity and high mortality rates if they are not managed properly. By applying AI technology, this study has developed an LSTM-based decision support system learning from patients' medical records and decisions made by clinicians so that it provides suggestions for the need for total health care. The proposed system can enhance the existing hospital information system with the ability to monitor the health condition of in-patients continuously and fill the communication gap among multiple clinic staff taking care of them. By allocating healthcare resources properly to the need, the proposed system can improve staff work efficiency as well as increase medical service quality.

The traditional rule-based CDSS failed to identify HHC needs and could not meet the expected benefit of the HHC service. It is critical to identify HHC needs to prevent worsening the patient's illness. Based on the experimental results, the proposed LSTM prediction model identifies HHC cases efficiently, yields high sensitivity, and hence is practical to real environments.

### Future work

This study built up an LSTM-based model to predict HHC needs. The developed clinical decision support system for HHC aims to integrate seamlessly into the existing hospital information systems, enhance the prediction of HHC needs, and improve healthcare quality and patient satisfaction.

## Author Contributions

**Conceptualization:** Wang-Chuan Juang, Ming-Hsia Hsu.

**Data curation:** Ming-Hsia Hsu, Zheng-Xun Cai.

**Funding acquisition:** Ming-Hsia Hsu.

**Investigation:** Zheng-Xun Cai, Chia-Mei Chen.

**Methodology:** Zheng-Xun Cai, Chia-Mei Chen.

**Project administration:** Wang-Chuan Juang.

**Software:** Zheng-Xun Cai.

**Supervision:** Wang-Chuan Juang, Chia-Mei Chen.

**Validation:** Wang-Chuan Juang, Ming-Hsia Hsu.

**Writing – original draft:** Zheng-Xun Cai, Chia-Mei Chen.

**Writing – review & editing:** Wang-Chuan Juang, Ming-Hsia Hsu, Zheng-Xun Cai, Chia-Mei Chen.

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
