## [Decision Letter · Decision Letter 0]

14 Jul 2022

PONE-D-21-31708Developing an AI-assisted Clinical Decision Support System to Enhance In-patient Holistic Health CarePLOS ONE

Dear Dr. Chen,

Thank you for submitting your manuscript to PLOS ONE. After careful consideration, we feel that it has merit but does not fully meet PLOS ONE’s publication criteria as it currently stands. Therefore, we invite you to submit a revised version of the manuscript that addresses the points raised during the review process.

We look forward to receiving your revised manuscript.

Kind regards,

Siuly Siuly, PhD

Academic Editor

PLOS ONE

Journal Requirements:

3. Thank you for stating the following financial disclosure: "Hsu has received the funding from  Kaohsiung Veterans General Hospital with grant number KSVNSU110-012.

Website: https://eng.vghks.gov.tw"

We note that one or more of the authors is affiliated with the funding organization, indicating the funder may have had some role in the design, data collection, analysis or preparation of your manuscript for publication; in other words, the funder played an indirect role through the participation of the co-authors. If the funding organization did not play a role in the study design, data collection and analysis, decision to publish, or preparation of the manuscript and only provided financial support in the form of authors' salaries and/or research materials, please do the following:

a. Review your statements relating to the author contributions, and ensure you have specifically and accurately indicated the role(s) that these authors had in your study. These amendments should be made in the online form.

b. Confirm in your cover letter that you agree with the following statement, and we will change the online submission form on your behalf: 

“The funder provided support in the form of salaries for authors [insert relevant initials], but did not have any additional role in the study design, data collection and analysis, decision to publish, or preparation of the manuscript. The specific roles of these authors are articulated in the ‘author contributions’ section.

5. We note that Figure 2 in your submission contain copyrighted images. All PLOS content is published under the Creative Commons Attribution License (CC BY 4.0), which means that the manuscript, images, and Supporting Information files will be freely available online, and any third party is permitted to access, download, copy, distribute, and use these materials in any way, even commercially, with proper attribution. For more information, see our copyright guidelines: http://journals.plos.org/plosone/s/licenses-and-copyright.

Reviewers' comments:

Reviewer's Responses to Questions

**Comments to the Author**

1. Is the manuscript technically sound, and do the data support the conclusions?

Reviewer #1: Partly

Reviewer #2: Yes

2. Has the statistical analysis been performed appropriately and rigorously? 

Reviewer #1: No

Reviewer #2: Yes

3. Have the authors made all data underlying the findings in their manuscript fully available?

Reviewer #1: No

Reviewer #2: No

4. Is the manuscript presented in an intelligible fashion and written in standard English?

Reviewer #1: Yes

Reviewer #2: Yes

5. Review Comments to the Author

Reviewer #1: The manuscript is of a great effort, particularly collecting patients heath data. However, it will be wonderful to address some pints mentioned below:

1. Abstract should be written in one or two paragraphs without the subsections. One is most preferred. It is the summary of entire manuscript.

2. There is no much contribution on proposed method. If the authors does not brings novelty in the method, at least make the comparison with at least four to five existing methods/algorithms.

3. The Performance matrix provided in the result section is not enough. The readers would like to see more analysis, like precision, recall, f1 measure etc.

4. The references are not enough. adding more state-of-the-art references would bring value to the manuscript.

Thank you.

Reviewer #2: It is very interesting to read your work on Developing an AI-assisted Clinical Decision Support System to Enhance In-patient Holistic Health Care. I found the study contributing to the body of knowledge. It wIll be interesting to read your further works.

6. PLOS authors have the option to publish the peer review history of their article (what does this mean?). If published, this will include your full peer review and any attached files.

Reviewer #1: **Yes: **Shapla Khanam

Reviewer #2: No

---

## [Author Response · Author response to Decision Letter 0]

17 Aug 2022

Reviewer #1

Thank you for the valuable suggestion, we have revised based on your suggestion accordingly.

Reviewer #2

Thank you for your comment. We will be honored to have our work published in the journal.

---

## [Decision Letter · Decision Letter 1]

10 Oct 2022

Developing an AI-assisted Clinical Decision Support System to Enhance In-patient Holistic Health Care

PONE-D-21-31708R1

Dear Dr. Chen,

We’re pleased to inform you that your manuscript has been judged scientifically suitable for publication and will be formally accepted for publication once it meets all outstanding technical requirements.

Kind regards,

Mohamed Hammad, Ph.D.

Academic Editor

PLOS ONE

Additional Editor Comments (optional):

Reviewers' comments:

Reviewer's Responses to Questions

**Comments to the Author**

1. If the authors have adequately addressed your comments raised in a previous round of review and you feel that this manuscript is now acceptable for publication, you may indicate that here to bypass the “Comments to the Author” section, enter your conflict of interest statement in the “Confidential to Editor” section, and submit your "Accept" recommendation.

Reviewer #2: All comments have been addressed

2. Is the manuscript technically sound, and do the data support the conclusions?

Reviewer #2: Yes

3. Has the statistical analysis been performed appropriately and rigorously? 

Reviewer #2: Yes

4. Have the authors made all data underlying the findings in their manuscript fully available?

Reviewer #2: No

5. Is the manuscript presented in an intelligible fashion and written in standard English?

Reviewer #2: Yes

6. Review Comments to the Author

Reviewer #2: (No Response)

7. PLOS authors have the option to publish the peer review history of their article (what does this mean?). If published, this will include your full peer review and any attached files.

Reviewer #2: No

---

## [Editor Report · Acceptance letter]

21 Oct 2022

PONE-D-21-31708R1 

Developing an AI-assisted Clinical Decision Support System to Enhance In-patient Holistic Health Care 

Dear Dr. Chen:

I'm pleased to inform you that your manuscript has been deemed suitable for publication in PLOS ONE. Congratulations! Your manuscript is now with our production department. 

Kind regards, 

on behalf of

Dr. Mohamed Hammad 

Academic Editor

PLOS ONE